# Analysis of the Factors Affecting Static In Vitro Pepsinolysis of Food Proteins

**DOI:** 10.3390/molecules27041260

**Published:** 2022-02-14

**Authors:** Natsumi Maeda, Dorota Dulko, Adam Macierzanka, Christian Jungnickel

**Affiliations:** Department of Colloid and Lipid Science, Faculty of Chemistry, Gdańsk University of Technology, ul. Narutowicza 11/12, 80-233 Gdańsk, Poland; s179071@student.pg.edu.pl (N.M.); dorota.dulko@pg.edu.pl (D.D.); adamacie@pg.edu.pl (A.M.)

**Keywords:** pepsinolysis, protein hydrolysis, half-time, additives, form comparison, in vitro digestion

## Abstract

In this meta-analysis, we collected 58 publications spanning the last seven decades that reported static in vitro protein gastric digestion results. A number of descriptors of the pepsinolysis process were extracted, including protein type; pepsin activity and concentration; protein concentration; pH; additives; protein form (e.g., ‘native’, ‘emulsion’, ‘gel’, etc.); molecular weight of the protein; treatment; temperature; and half-times (HT) of protein digestion. After careful analysis and the application of statistical techniques and regression models, several general conclusions could be extracted from the data. The protein form to digest the fastest was ‘emulsion’. The rate of pepsinolysis in the emulsion was largely independent of the protein type, whereas the gastric digestion of the native protein in the solution was strongly dependent on the protein type. The pepsinolysis was shown to be strongly dependent on the structural components of the proteins digested—specifically, β-sheet-inhibited and amino acid, leucine, methionine, and proline-promoted digestion. Interestingly, we found that additives included in the digestion mix to alter protein hydrolysis had, in general, a negligible effect in comparison to the clear importance of the protein form or additional treatment. Overall, the findings allowed for the targeted creation of foods for fast or slow protein digestion, depending on the nutritional needs.

## 1. Introduction

Proteins are an important ingredient in the human diet. They are involved in muscle growth, immune function, hormone synthesis, tissue repair, and other physiological functions. The estimated dietary protein requirement for adults is 0.7–1.0 g/kg body weight per day [1,2,3]. Humans consume proteins in a plethora of different forms (e.g., meat, dairy, eggs, supplements, etc.) and of different origins (e.g., plant or animal proteins). Most sources of food proteins consist not only of pure protein but also of other nutrients and bioactive components (e.g., carbohydrates, lipids, polyphenols, vitamins, etc.). Those components, among other factors such as protein concentration, structure, or molecular weight, can affect the digestibility of dietary proteins by human gastrointestinal enzymes [4,5,6], often with contrasting effects. For example, polyphenol extracts were shown to delay in vitro pepsin digestion of milk protein β-lactoglobulin (BLG) owing to aggregates formed by strong noncovalent interactions between the protein and polyphenol [7]. A different study used the same protein with the addition of phosphatidylcholine for simulated gastric proteolysis [8]. The lipid did not impact gastric digestion, while it showed a protective effect during the following duodenal proteolysis of BLG. Micro- and macro-structural organizations of proteins in foods are often generated by various food-processing methods (e.g., emulsification, heating, gelation, and enzymatic treatment). Although these are necessary for creating functional structures in foods, the processing methods can alter the protein digestibility and impact the availability of amino acids. As an example, studies using in vitro digestion models have shown that an emulsification process can significantly increase the rates of gastric digestion of BLG and β-casein at pH 1.5–2.5 [9,10,11], due to the conformational changes of the proteins at the oil/water interface caused by adsorption and the consequent changes in the flexibility of the proteins, which allowed pepsin to cleave them more efficiently. However, a reverse effect in β-casein-stabilized emulsion was also shown more recently [12]. The major difference between that study and the studies mentioned above was that the in vitro proteolysis was carried out at pH 5.5 to reproduce the early stage of gastric digestion. This might have impacted the enzyme activity and the protein substrate and accounted for the retarded pepsinolysis observed. Another example of food processing that can affect the digestibility of proteins is gelation. In particular, heat-induced protein gel formation is extremely complex and can involve a combination of several types of reactions, including denaturation, dissociation–association, and aggregation. The heat-induced gelation of dietary proteins is commonly used in food manufacturing to provide a desired texture, viscoelasticity, and stability of a vast range of food products by the formation of three-dimensional gel networks [13,14]. Nevertheless, relatively little is known about how the gelation conditions influence the digestibility of dietary proteins. The study reported by Macierzanka et al. [15] can be used as an example for contrasting rates of digestion. In that study, BLG was exposed to a range of thermal and pH conditions to form stranded or particulate gels. Heating the protein to 85 °C for 30 min at pH 5.2 formed particulate gels with a greater overall resistance to proteolysis than heating to 70 °C for 24 h. A comparison with a range of pH values was also investigated in the study where the gel produced at pH 6.5 showed the highest degradation of the protein during digestion, followed by pH 2.5, pH 5.2, and pH 4.8.

The gastric digestibility of proteins is very complex and often exhibits contrasting effects depending on many factors, such as protein type and colloidal form, pH, additional treatment, etc. However, this is important research, especially in the rapidly developing sector of functional foods, due to the growing interest in engineering properties of dietary proteins for modulating digestion and improving nutrient release in the gastrointestinal tract and, ultimately, for controlling the physiological response in individuals with different characteristics (e.g., health status, age, etc.).

Our objective, therefore, was to identify an overall pattern of the importance of various factors in the pepsinolysis of dietary proteins (e.g., protein form, protein concentration, pepsin concentration, pH, molecular weight of protein, etc.). For that purpose, we determined the statistical significance of the factors for pepsinolysis using the scientific literature data available from the last 70 years, conducted statistical analysis with the quantitative pepsinolysis data, and used a protein HT (the time at which 50% of the protein digestion was completed) as dependent variables.

## 2. Results and Discussion

The proteins were classified into five protein forms. The first is ‘native’ proteins; this group includes proteins dissolved in an aqueous solution before digestion. The second is protein-stabilized ‘emulsion’; the dispersion of two or more immiscible liquids (e.g., water and oil) where the protein is adsorbed at the interface, lowering the interfacial tension and forming protective membranes around the emulsion droplets. The third protein form is ‘gel’, where physical or chemical methods are used to denature or cross-link protein molecules, which results in the formation of gels. The fourth form is the proteins in ‘real food’ (e.g., meat and peanuts), which is a solid or semi-solid food matrix consisting of different nutrients, including naturally existing proteins and not commercially available and/or additionally incorporated proteins, such as protein powder supplements. ‘Milk’ has been classified as a separate, fifth protein form. This difference is clear when analyzing the dominant bovine milk in our analysis, as it neither represents just protein adsorbed onto the oil–water interface in ‘emulsion’ or just a ‘native’ protein or a ‘real food’ in the context of this analysis. To look at the milk issue in more detail first, the constituents of milk should be mentioned. Using bovine milk as an example, milk has a complex composition (Figure 1) consisting of milk fat globules that are surrounded by surface-active lipids—predominantly phospholipids organized in the form of a tri-layer membrane—and some proteins (this lipid-and-protein structure is referred to as the milk fat globule membrane (MFGM)), casein micelles consisting of αs1, αs2, β, and κ-casein, as well as calcium phosphate [16], and the serum containing soluble whey protein, mainly BLG and α-lactoglobulin (ALA) [17]. Milk presented some difficulty to categorize as a ‘real food’, because it contains emulsified milk fat globules. It has not been classified as ‘emulsion’, as the fat globules are stabilized predominantly by adsorbed phospholipids, which is different than in a protein-stabilized ‘emulsion’ (Figure 1B). The presence of native whey proteins might allow the milk to be classified as a ‘native’ form of protein, although the whey proteins only represent a minor fraction of the total protein milk content, which is dominated by caseins organized in complex micellar nanostructures [17]. It should also be noted that infant milk formulas (IMFs) have been categorized as ‘emulsion’ for this study. Conventional IMFs consist of lipid droplets stabilized by proteins adsorbed to the droplets’ surface [18], whereas fat droplets in real milk are stabilized by phospholipid-dominated tri-layers, as explained above.

The partial least squares regression (PLS) model was used here to determine the variable importance score, which summarizes the contribution each variable makes to the model. It is determined as a weighted sum of the square correlations between the descriptors and the dependent variable. The results of the PLS analysis are shown in Figure 2. Numerically, the PLS fit has the following error statistics, MAE (Mean Absolute Error): 55.18, RMSE (Root Mean Squared Error): 95.2, and R-squared: 0.147. Variable importance in the projection (VIP) (Figure 2A) indicates that the protein forms of ‘native’, ‘milk’, ‘emulsion’, pH, and molecular weight (MW) of a protein are the dominant factors influencing protein HT during pepsinolysis. Interestingly, the additive had the smallest impact on the HT. An ‘additive’ refers to any substance (e.g., polyphenol and mucin) that was added to the digestion mix to affect the protein digestibility. The range of values for which the calculations are valid is shown in Table 1. The effect on the rate of protein hydrolysis can be separated into several contributing factors, which will be described below.

### 2.1. Effect of Protein Type

The majority of proteins for which the pepsinolysis data have been extracted from the literature were BLG, ALA, casein, and MFGM proteins, consisting of a total of 57% of all measured HT. The results of the PLS analysis for different types of proteins are shown in Figure 3A,B. The VIP-illustrated BLG and MFGM had a significant impact on HT, followed by the casein and ALA comparably small impacts (Figure 3A). The standardized coefficient illustrated that only BLG had a negative impact on HT. The key factor influencing the protein susceptibility to pepsin is the protein structure, while the structure will differ depending on protein type, protein form, pH, temperature, etc. BLG is known to be resistant to proteolysis by pepsin at low pH, as it forms a compact stable structure [25,26] by creating internal hydrogen bonds, which might inhibit the access of pepsin to potential cleavage sites in the folded calyx structure of β-strands [27]. In addition, it is known that pepsin preferentially hydrolyzes peptide bonds between aromatic amino acids (Phe, Trp, and Tyr); Leu; and Met residues [28]. The number of aromatic amino acid groups, Leu, and Met per molecule of protein is given in Table 2. It can be seen that BLG, which has the highest resistance to pepsinolysis, also has the smallest percentage of aromatic amino acid, Leu, and Met residues (22.3%), while ALA with 27% might be more susceptible. Not only is the absolute number of aromatic amino acid, Leu, and Met residues important but also, the secondary and tertiary structures of the proteins and their ability to denature. Specifically, secondary structure denaturation happens when α-helices and β-sheets lose their shape, where, especially, the β-sheet is more responsible for the rigidity of the protein form [29]. Tertiary structure denaturation occurs when bonds for three-dimensional structures are disrupted, e.g., disulfide bonds. The presence of proline amino acids disrupts the secondary structure, as it is too rigid to be incorporated into secondary structures [30]. These residues, therefore, present a potential point of action of the pepsin. Caseins have a smaller percentage of β-sheets and more proline residues compared to BLG (Table 2), which makes caseins’ structure nonrigid. In terms of tertiary structure, caseins contain a low disulfide bond content, making caseins more nonrigid on a tertiary level [31], in addition to the primary and secondary structures. Therefore, considering these three structural aspects of caseins, they experience faster digestion. In regard to MFGM proteins, likewise, the number of aromatic amino acid residues, Leu, and Met in MFGM proteins that pepsin is supposed to cleave preferably is significantly higher than in any other protein types (Table 2). When analyzing the pepsinolysis data in terms of the structural components (Figure 4A,B), these trends are confirmed, i.e., aromatic amino acid residues, leucine, and methionine, and proline residues strongly encourage the digestion of protein, and the rigid structure of the β-sheet reduces digestibility. This comparison was also conducted for intact (groups ‘native’ and ‘milk’) and restructured (groups ‘gel’ and ‘emulsion’) proteins, as shown in Figure 4C–F, where the effect of the secondary structural components was more pronounced in the intact proteins when compared to restructured forms (‘restructured’ is used instead of ‘denatured’ to allow the inclusion of folded or partially unfolded proteins in some emulsions and gels, where protein unfolding was not clear from the information provided). In addition, when analyzing the properties of proteins, specifically, the net charge and hydrophobicity were analyzed (variables with a VIP > 1). It was found that the net charge had a statistically significant negative contribution to the HT of protein digestion, meaning the higher the net charge, the faster the rate of digestion, due to the stronger aspartate–substrate interaction (the active site in pepsin is comprised of two aspartate residues). The net charge of the proteins increases with the lowering pH, which also coincides with the increased hydrolytic activity of pepsin. The enzyme has been reported to have the highest activity at pH around 2.5 [32], and at this pH, dietary proteins usually exhibit a substantial positive net charge. Hydrophobicity was also found to have a similar negative contribution. The importance of the net charge and hydrophobicity, however, declined for restructured proteins, as the restructured structures of the proteins are often no longer sterically hindered from entering the aspartate active site of the pepsin, and thus, the strong positive net charge is not required.

### 2.2. Effect of Protein Form

When analyzing the protein forms, it is evident that the ‘milk’ form has the shortest pepsinolysis HT (Figure 2B), and therefore, the protein in ‘milk’ is broken down the fastest, followed by the protein in ‘emulsion’. This is clear when comparing the results to Figure 3, as research groups either tested bovine milk [46] or concentrated casein protein [47]. The bovine milk protein fraction is comprised of approximately 33:8:1 casein:whey:MFGM [45,48], and it was shown before caseins and MFGM proteins are hydrolyzed with ease by pepsin.

Emulsions, independent of their protein type (*p* > 0.05), showed a fast pepsinolysis HT (Figure 2B), most likely due to the restructured conformation of the protein at the interface caused by adsorption [49] and the large surface area of the emulsion droplets. On the other side of the scale, native and real food have the longest pepsinolysis HT. ‘Native’ presents the lowest standardized coefficient, indicating that, without restructuring the protein, the rate of proteolysis is slow. However, it should be noted that this result is due to the dominance of the studies on the digestibility of dissolved proteins being conducted with whey proteins. BLG was the most frequently tested protein type, consisting of around 30% of our gathered data set, thus the properties of BLG contributed to the result. The pepsin resistance of native BLG was explained earlier in the manuscript.

‘Real food’ proteins also had a long HT, presumably due to the complex cellular matrix in which the proteins are present (e.g., proteins in plant or animal proteins). Although the result was not shown as a significant factor for pepsinolysis (Figure 2A), the comparison between heat-induced gelation and non-heat-induced gelation indicated a distinct difference in gastric digestion HT. The average HT of heat-induced BLG gelation was around 49 times faster than that of non-heat-induced BLG gelation, which made the overall digestibility of the protein gel fast (Figure 2B).

Comparison of the structural components of the restructured protein forms (‘emulsion’ and ‘gel’) with intact protein forms (‘native’ and ‘milk’) (Figure 4C–F) showed a reduction of the importance of β-sheets related to the form of the protein.

### 2.3. Effect of pH

pH was shown to be the third-most significant descriptor, with a negative effect of pH on the protein HT during pepsinolysis, indicating that a higher pH promoted faster digestion (Figure 2). This was an unanticipated result, as the optimum pepsin activity is known to be around 2.5 [32]. It should be noted that our range of study was between pH 1.2 and 5.5 (Table 1), yet the analyses carried out at pH ≥ 4.0 were all caseins (Appendix A). As it was previously outlined, caseins are prone to fast pepsinolysis, which results in the significant negative effect of pH on the protein HT in our overall study. To confirm this, the PLS analysis was repeated without the pH ≥ 4.0 data points, and the standardized coefficient decreased from −0.111 (with pH ≥ 4.0) to −0.095 (without pH ≥ 4.0). BLG (the most frequently studied protein; 30% of all data points), at pH below 2 (54% of BLG data points) presents a compact structure with a smaller volume, retarding the access of pepsin [50]. However, no studies with BLG above pH 3, which would show the retardation of pepsinolysis due to dimer and octamer formation, were conducted. Additionally, studies with a pH value above 3.5 were mostly conducted with emulsions. A Kruskal–Wallis test of pH to protein form indicated that the pH used for the assays of ‘emulsion’ was significantly different from the rest of the protein forms (*p* > 0.05). As stated previously, ‘emulsion’ exhibited the fastest digestion rate according to our results. 

### 2.4. Effect of Treatment

Here, ‘treatment’ is defined as a physicochemical method that restructured the protein but was neither categorized as an additive or processing that changed protein form, i.e., heating for creating gels or dispersing for creating emulsions (and, thus, excluded from the ‘treatment’ group). The ‘treatment’ category was then further divided into physical or chemical treatments. The results showed that both chemical and physical treatments did not have a significant effect on pepsinolysis, with low variable importance (Figure 2A). It was noted that the chemical treatment had a higher impact on HT than the physical treatment, while the presence of both treatments decreased the HT of pepsinolysis (Figure 2B). 

Even though heat treatment was the most common ‘treatment’ method studied (64% of all treatment cases), the small effect of physical treatment on the HT was reduced by the wide range of temperatures (as these were not modeled separately) used for treatment (ranging between 50 and 140 °C, with an average of 88 °C) and a contrasting range of effects. This resulted in a diminished significance of the variable. For example, in one case, BLG and ALA from infant milk formula were heated to 67.5 °C or 80 °C, resulting in no effect on the rate of pepsinolysis [51]. However, a different research group heated BLG to 90 °C (for 5 or 120 min) and showed a significant increase in HT [52]. This is despite the fact that the heat-induced unfolding of BLG has occurred in a broad range of 60–90 °C [53]. Analyzing the ‘real food’ form of protein with meat protein, Bax et al. [54] showed the different accessibilities of pepsin to protein following different conformational changes of the protein at varying temperatures. Specifically, at 70 °C, protein denaturation occurred by exposing the hydrophobic part of the protein, which rendered the protein more digestible. Above 100 °C, the heat treatment promoted aggregation by chemical reaction, i.e., oxidation or deamination, which inhibited the access of pepsin to cleavage sites of the protein.

In terms of chemical treatment, it has to be noted that only one kind of chemical treatment was included, which applied β-mercaptoethanol (BME) and urea. Both chemicals are known as protein-denaturing agents, where BME breaks disulfide bonds [55] and urea denatures protein by a couple of mechanisms such as the formation of hydrogen bonds with polar residues containing amino acid side chains and the disruption by a hydrophobic collapse [56]. Although more papers describing chemical treatments would lead to more reliable outcomes, our analysis showed the impact of one chemical treatment over different proteins, and the significance of the mentioned chemical treatment was higher than that of the physical treatment. Therefore, a treatment that does not change the protein ‘form’ but only serves the purpose of restructuring the protein has a lower impact on pepsinolysis than processing, which changes the protein ‘form’ (e.g., emulsion, gel, etc.). 

### 2.5. Importance of Additives

The importance of ‘additives’ on proteolysis by pepsin was surprisingly the smallest among all the other categories (Figure 2A). Examples of the effect of additives include sodium bisulfite (NaHSO_3_), utilized as a preservative [57], which promoted fast pepsinolysis of the sorghum protein by disrupting the intermolecular disulfide linkages of protein polymer-bound through cleavage and thus allowed the proteins to be extracted [58]. Conversely, commonly cited polyphenol extracts (containing catechins and flavonoids) from green tea, black tea, cocoa, and coffee were shown to delay the pepsinolysis of BLG due to the aggregate formation between the protein and a polyphenol caused by their noncovalent interactions [7]. Therefore, even though these individual effects presented by the researchers are significant and important, the overall rate of pepsinolysis may be altered much more significantly by changing the form of the protein. However, if changing the form is not an option, functional foods might be generated with the use of additives to change the protein digestibility. It should be noted, however, that the changes induced by additives allow for HT changes up to 94% (e.g., for ‘gels’) from those without an additive for a given form of the protein. As shown in Figure 5, it can be seen that additives have more significant effects on different protein forms, specifically on the ‘real food’ and ‘native’ protein forms, since these are two groups that represent proteins in their intact form (meaning not restructured by, for example, gelation or emulsification). 

### 2.6. Effect of Molecular Weight, Pepsin, and Protein Concentration

The MWs of all the proteins presented a relatively significant effect on pepsinolysis above the variable importance (Figure 2A). The result of the standardized coefficient illustrated that a higher MW of the protein within the studied range (Table 1) resulted in faster hydrolysis of the protein by pepsin (Figure 2B). All the large MW proteins (caseins and MFGM) are also easily hydrolyzable, the result is not surprising (as they present more contact sites [59,60]). Similar to MW, the concentration of protein promotes faster digestion. The pepsin and the protein concentrations were placed third and second-least important, respectively (Figure 2A), showing only a slight impact on pepsinolysis. 

### 2.7. Possible Improvement for Future Research—Irregular Pepsin Concentration, and Reported pH

While conducting this research, it has come to our attention that the authors did not use consistent conditions to conduct the in vitro digestion experiments. Previous attempts have been made to standardize the methods used for digestion, most notably by the INFOGEST network [61,62]. However, the conditions used to determine protein digestibility are noticeably different. Firstly, we encountered inconsistent enzyme activity units and insufficient reporting of pepsin or protein concentrations and pepsin activity. Due to the lack of information of pepsin activity, our study only included the pepsin concentration. However, in Table 3, we show the Kruskal–Wallis test of the ratios of pepsin concentration to the amount of protein present versus the ‘form’ of the protein, where clearly the effects are statistically different. This highlights the need for consistent experimental conditions, as suggested by Minekus et al. [63], and/or for giving justification if conditions are different than those recommended.

Lastly, the point of pH measurement of the pepsinolysis assay was not consistent among the papers analyzed. In the course of our research, we noted that 63% of the authors provided the pH value at the stage where digestion starts (i.e., after a protein preparation has been mixed with a simulated gastric fluid, SGF), 21% provided only the pH value of SGF, and 16% provided the value of both. To have comparable results, the measurement point should be ideally consistent and provide the pH of the digestion mix. The INFOGEST in vitro digestion protocol suggests a pH 3.0 of the protein mixed with SGF for digestion [61].

## 3. Methods

### 3.1. Data Collection and Extraction

Data collection was carried out using Google Scholar and Thomson Reuter’s Web of Science. Google Scholar covers scientific information from a broad range of fields available online that is independent of the journal [64]. Web of Science is one of the richest sources of journals, with more than 8700 journals from the year 1900 onwards [65]. Therefore, a combination of the two databases should provide the majority of all available papers [66]. The search terms “protein digestion pepsin”, “protein half-life”, and “pepsinolysis” were used. Only those papers that presented quantitative data on the pepsinolysis rate (i.e., the rate of decrease in protein concentration due to hydrolysis by pepsin) were included in this study. Fifty-eight papers were found from the two databases, with the earliest paper from 1956 to the latest from 2020.

Eleven descriptors were extracted from the papers—specifically, HT (min), protein name and form, additives, protein concentration (mg/mL), pepsin activity and/or concentration (mg/mL), pH, molecular weight of protein, temperature, and protein treatment. Papers that did not record or allow for a calculation of a proteolysis HT and protein concentration were not included in the study. Moreover, in vivo or (semi-)dynamic in vitro gastric digestion studies were omitted, as they are intrinsically not comparable to the predominant in vitro static studies (i.e., with constant pH during the digestion). Consequently, 143 unique datasets from 35 papers, spanning 1991–2020, were included in this analysis. The complete data is shown in Appendix A.

In the analysis, the proteolysis HT was set as the dependent variable. The HT quantifies the time at which half of the initial quantity of the protein is hydrolyzed by pepsin under gastric conditions. Proteins were classified into five forms—specifically, ‘native’ protein, protein-stabilized ‘emulsion’, protein ‘gel’, ‘milk’, and protein in ‘real food’. Details about the protein form were described in Section 2 (Results and Discussion). 

Using the Kruskal–Wallis test to compare (as shown in Table 4) the HT of milk with the other categories, we found that milk is significantly different from ‘emulsion’, ‘real food’, and the ‘native’ protein form, indicating that milk cannot be classified as any of those categories in terms of the protein form. Therefore, ‘milk’ was classified as a separate protein form in any further analysis. It should be noted here that this does not extend to conventional infant milk formula (IMF), which is considered a protein-stabilized emulsion, as explained in Section 2 (Results and Discussion).

Additives were defined as added substances that interact with the protein and affect its structure to either enhance (designated as +1) or inhibit (designated as −1) the rate of pepsinolysis. This did not include additives that were added to change the protein forms mentioned above, e.g., an oil phase in the emulsion or enzymes for promoting protein gelation. Protein and pepsin concentrations were recalculated to use the same unit, mg/mL. Details for the recalculation are given in the Appendix A. The pH was recorded either for the simulated fasted state, i.e., of the simulated gastric fluid (SGF) or for the beginning of gastric digestion (i.e., after the mixing of a protein preparation with SGF). The exact point of the pH measurement was recorded. The MW of single proteins was extracted directly, or, if not listed, it was taken from UniProt [43]. The temperature of the digestion experiments was uniformly at 37 °C and was therefore not included in further analyses. The additional ‘treatment’ category was designated as a treatment before pepsinolysis, which restructured the protein but was neither categorized as an additive or changed in protein form, as mentioned above (e.g., heating for creating gels or mixing for creating emulsions). This category was further divided into physical or chemical treatment. The structural components of proteins were collected and calculated from the literature, and, if not available, were taken from UniProt [43]. Net-charge (at pH 3.0) and hydrophobicity of the proteins were calculated using R [67], with the package ‘peptide’ [44]. Unfortunately, it was not possible to include pepsin activity in the list of descriptors, as 23% of the data from the papers did not include the activities. Moreover, 77% of the provided pepsin activity was expressed in different units (e.g., U/mg of protein, Unit, protein, U/mL of SGF, FIP-unit/mg, etc.), and straightforward conversion of the units was not possible. Since only 36% of the data had the same pepsin unit (U/mg of protein) overall, it was not prudent to estimate the remaining 64% with Multivariate Imputation by Chained Equations (MICE) either. The methods used to recalculate the descriptors, including the SDS-PAGE gel band intensities, are provided in the Appendix A. The workflow of the data collection, classification, and analysis has been depicted in Figure 6.

### 3.2. Numerical Techniques

When analyzing the 143 data points, a total of 8.8% of the data was noncontinuous, meaning that some of the information (most commonly the pepsin concentration) was missing. Since the amount missing was less than 10%, it was decided to use Multivariate Imputation by Chained Equations (MICE) to estimate the missing data. The MICE method is described in detail in Reference [68]. In brief, each missing value was estimated, in turn, based on its distribution, using regression models. Each missing variable is then recalculated n-times based on new predictions. It was found that six imputations resulted in the least normalized root mean square error and were thus applied, as shown in Appendix A. 

Partial least squares (PLS) regression [69] was used to determine the descriptor that has the most significant influence on protein digestion. The XLSTAT implementation of PLS regression was used [70,71]. For cross-validation, the jackknife leave-one-out method was used. The variables were centered and reduced before modeling. The variables’ importance for the projection (VIPs) was determined by Equation (1), according to Farres et al. [72]:(1)$$VIP_{j}=\sqrt{\frac{\sum_{f=1}^{F}w_{jf}^{2}\cdot SSY_{f}\cdot J}{SSY_{\mathrm{total}}\cdot F}}$$
where *w_jf_* is the weight value for the *j* variable and *f* component, and *SSY_f_* is the sum of squares of the explained variance for the *f*th component and *J* number of *x* variables. *SSY*_total_ is the total sum of squares explained by the dependent variable, and *F* is the total number of components.

Kruskal–Wallis test with post-hoc Dunn analysis was applied to determine the significance, since the observations were the result of different research groups and can, therefore, be considered independent. 

## 4. Conclusions

In this work, we analyzed the pepsinolysis of food proteins and the various factors that affect them. We collected papers from 1956 to 2020 and meticulously extracted and analyzed the techniques and results reported in the papers. Two statistical methods were used to compare protein groups: the Kruskal–Walls test and partial least squares (PLS) regression.

We found that the protein form is the most important determinant that influences the rate of pepsinolysis, with both ‘milk’ and ‘emulsion’ forms that led to the fastest pepsinolysis, followed by ‘real food’ and ‘gel’ forms that yielded slower rates of digestion. The pH had an interestingly counterintuitive effect on the HT of protein hydrolysis, resulting from the compaction of BLG (as the most measured protein) at lower pH values. Furthermore, additional additives used to promote or inhibit pepsinolysis, as well as methods of additional physicochemical treatment, have limited effect. However, the greatest effect for additives was shown in the ‘real food’ and ‘native’ protein forms. We clearly demonstrated the number of available aromatic amino acid, leucine, methionine, and proline residues in a protein; the rigidity of the protein (as characterized by the β-sheet content), as well as the net charge of the protein, affect the pepsinolysis rate.

The collected papers also presented a range of inconsistent units and a lack of standard digestion conditions, indicating and restating the need for consistent methodologies and reporting, as indicated by Minekus et al. [63]. 

It has been possible to replicate major findings of other research groups using a large, collected set of data, which covers a wide variety of protein types and forms.

The work might help food engineers to use targeted techniques to design personalized foods, which would have fast or slow protein digestion rates. 

## Figures and Tables

**Figure 1 molecules-27-01260-f001:**
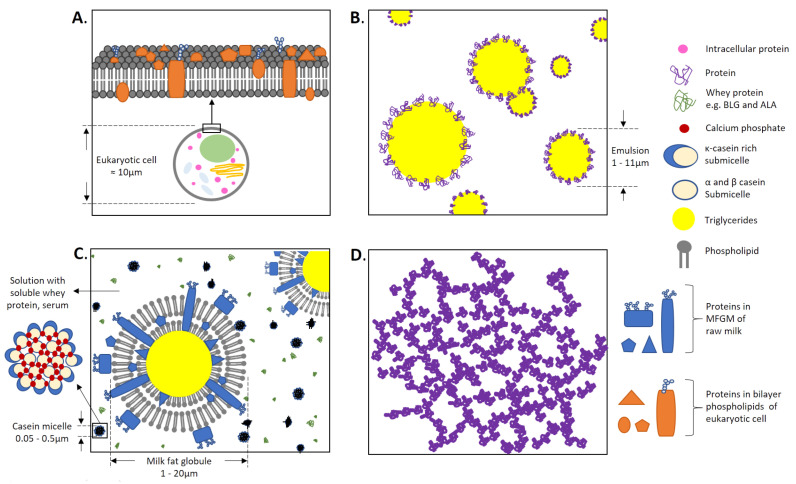
Schematic representation of four protein forms (**A**) ‘real food’, (**B**) ‘emulsion’, (**C**) ‘milk’, and (**D**) ‘gel’. Exemplary ‘native’ protein is visible as the green whey protein in (**C**). The issue of ‘milk’ classification is evident from the analysis of (**C**). The phospholipid bilayer with the transmembrane and peripheral proteins are evident in both (**A**,**C**). However, ‘milk’ also presents protein as micelles and native protein. The diameters were adapted from literature (**A**) [19], (**B**) [20], and (**C**) [21]. Casein micelle and milk fat globules were adapted from Horne and Singh & Gallier [22,23].

**Figure 2 molecules-27-01260-f002:**
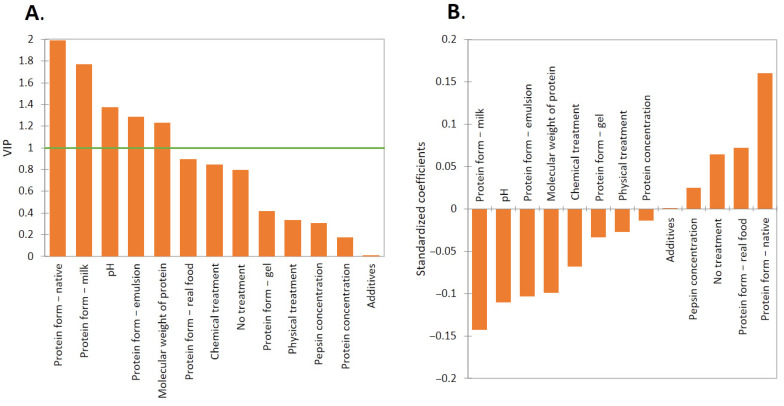
(**A**) Results of the PLS regression of the 143 pepsinolysis HT data points. Variables of importance in the projection (VIP) cut-off are marked in green. It can be seen that the type of additive has a marginal effect on the HT of pepsinolysis. (**B**) The ‘native’ and ‘milk’ protein forms have the largest impact on pepsinolysis, albeit in opposite directions, as indicated by the standardized coefficients (regression coefficients standardized, so that the variance of the independent variable is equal to 1). The ‘milk’ form results in the shortest HT, while the ‘native’ protein form results in the longest HT.

**Figure 3 molecules-27-01260-f003:**
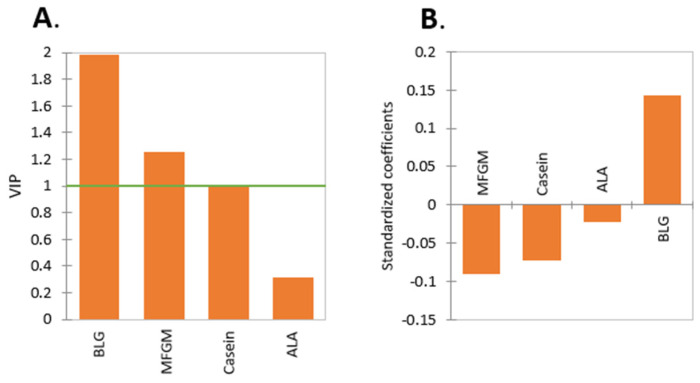
Comparison of the coefficients of PLS regression of four types of proteins. (**A**) The VIP of the four proteins, and (**B**) the standardized coefficients. It is clearly evident from (**B**) that BLG and ALA have the longest HT, while casein and MFGM protein have a shorter HT.

**Figure 4 molecules-27-01260-f004:**
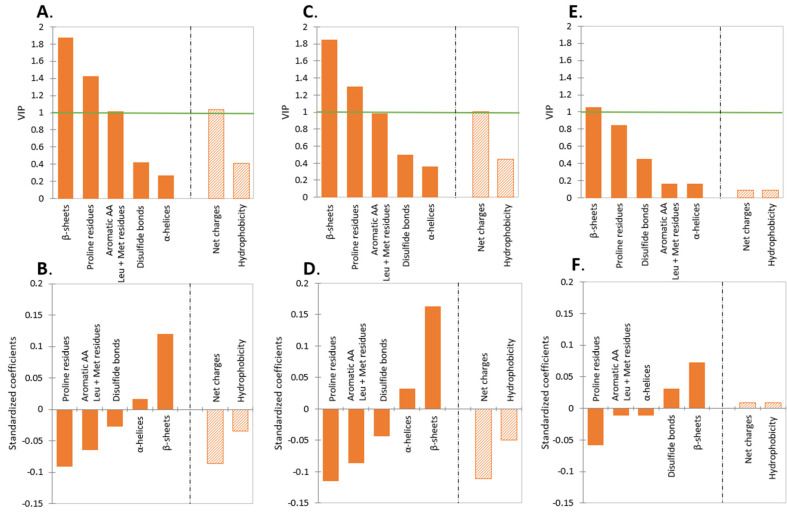
(**A**,**B**) The overall (i.e., for intact + restructured proteins) comparison and importance of the impact of the structural components of proteins on the pepsinolysis. (**C**,**D**) Protein in intact forms (groups ‘native’ and ‘milk’). (**E**,**F**) Proteins are in restructured forms (groups ‘gel’ and ‘emulsion’). For publications in which the casein composition was not specified, the ratio of casein constituents was calculated from the composition of bovine, equine, or human milk [45] where applicable. Hatched columns represent parameters from a separate calculation (as hatched and solid columns represent variables with overlapping physicochemical significance).

**Figure 5 molecules-27-01260-f005:**
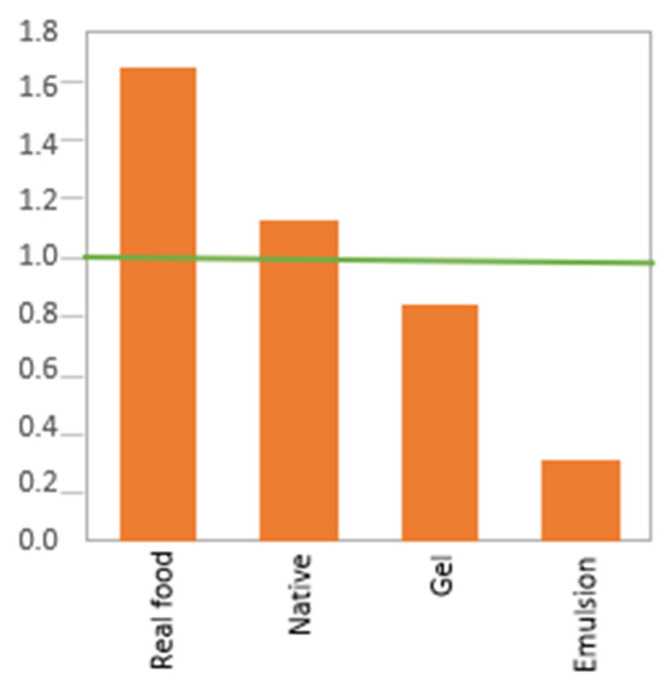
Comparative PLS analysis showing that additives have varying impacts on proteins presented in different forms, most significantly the ‘real food’ and ‘native’ protein forms.

**Figure 6 molecules-27-01260-f006:**
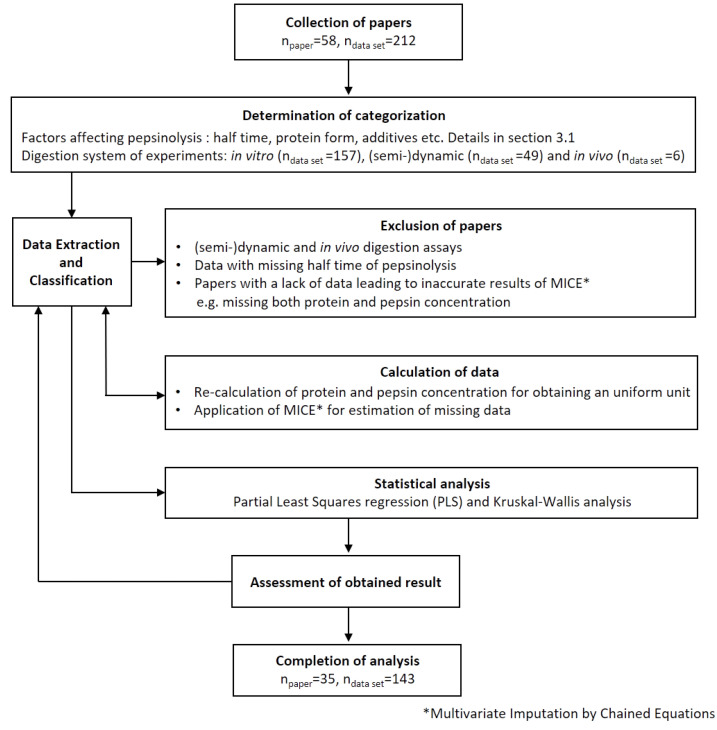
Flowchart presenting the workflow.

**Table 1 molecules-27-01260-t001:** Range of values for the descriptors collected from the literature. The minimum and maximum represent the boundary box of the applicability domain of the regression models according to Grammatica [24].

	Minimum	Maximum	1st Quartile	Median	3rd Quartile	Mean	Standard Deviation (n)
Additives	−1	1	0.00	0	0.00	0.02	0.51
Molecular weight (kDa)	5.80	270.00	18.00	24.00	51.11	41.18	44.47
Protein concentration (mg/mL)	1.00 × 10^−3^	54.25	0.25	0.60	1.00	2.09	5.44
Pepsin concentration (mg/mL)	5.00 × 10^−5^	3.05	0.24	0.80	1.42	0.84	0.66
pH	1.20	5.50	1.55	2.00	2.50	2.14	0.90
HT (min)	0.50	630.00	4.65	17.80	73.50	57.57	104.19

**Table 2 molecules-27-01260-t002:** Summary of the structural components of the major types of milk proteins. Shown are the aromatic amino acid (AA) residues (containing phenylalanine, tryptophan, and tyrosine) plus leucine and methionine residues, as well as the number of proline residues, which act as structural disruptors [33]. The numbers of amino acid residues were taken from literature ^a^ [34], ^b^ [35], and ^c^ [36]. The percentage composition of the beta-sheets and alpha-helices is derived from literature ^d^ [37], ^e^ [38], ^f^ [39], ^g^ [40], and ^h^ [41], and the number of disulfide bonds in the tertiary structure is from literature ^i^ [42]. Common MFGM proteins xanthine oxidase, lactadherin, and butyrophiline were used. If no specific reference was given, the data was supplemented from UniProt [43]. Protein properties were calculated in R using the ‘peptide’ package [44]. Protein sequences were taken from the sources indicated.

	Primary Structure			Secondary Structure		Tertiary Structure	Properties	
	Aromatic AA Residues + Leu and Met	Proline Residues	Total Residues	β-Sheet (%)	α-Helix (%)	Disulfide Bond (Excluding Free Cys)	Net-Charge (at pH 3)	Hydrophobicity
BLG	36 ^a^	8 ^a^	44	50 ^f^	15 ^f^	2 ^a^	18.5	0.037
ALA	24 ^b^	2 ^b^	26	14 ^g^	26 ^g^	4 ^b^	9.5	0.041
αs1-casein	42 ^c^	17 ^c^	59	20 ^h^	13 ^h^	0 ^c^	23.2	−0.052
αs2-casein	37 ^c^	10 ^c^	47	30 ^e^	24–32 ^e^	1 ^i^	31.5	−0.17
β-casein	43 ^c^	34 ^c^	77	22 ^h^	13 ^h^	0 ^c^	19.0	0.06
κ-casein	24 ^c^	20 ^c^	44	10 ^d^	30 ^d^	0 ^c^	16.5	−0.00042
Xanthine oxidase	261	71	332	25	32	1	166.9	0.038
Lactadherin (PAS VI/VII)	90	21	111	20	3	9	44.3	0.064
Butyrophilin (BTN1A1)	111	37	148	24	2	2	61.1	0.024

**Table 3 molecules-27-01260-t003:** Kruskal–Wallis analysis shows statistical significance for the comparison of the pepsin concentration to protein ratio at the start of digestion. Clearly evident is that pepsin ratios are different depending on which ‘form’ of protein is being analyzed.

	Emulsion	Gel	Milk	Native	Real Food
Emulsion	1	0.001	<0.0001	<0.0001	<0.0001
Gel		1	0.000	0.170	0.057
Milk			1	0.009	0.075
Native				1	0.443
Real food					1

**Table 4 molecules-27-01260-t004:** Kruskal–Wallis comparison of the HT of various protein forms. In bold is shown the comparison specifically to milk:emulsion, milk:native, and milk:real food. It is evident that the protein form ‘milk’ is significantly different (*p* > 0.05) to ‘emulsion’, ‘native’, and ‘real food’. Kruskal–Wallis test with post-hoc Dunn analysis was applied.

	Emulsion	Gel	Milk	Native	Real Food
Emulsion	1	0.079	**0.010**	0.300	0.001
Gel		1	0.333	0.004	<0.0001
Milk			1	**0.000**	**<0.0001**
Native				1	0.012
Real food					1

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
