# Peer review of "Analysis of the Factors Affecting Static In Vitro Pepsinolysis of Food Proteins"

_molecules, 2022, doi:10.3390/molecules27041260_

Round 1

Reviewer 1 Report

In this manuscript, the authors conducted a meta-analysis of 58 reports on the results of static in vitro protein gastric digestion in the past 70 years. It mainly includes protein type, pepsin activity and concentration, protein concentration, pH value, additive, protein form (such as "natural", "emulsion", "gel", etc.), protein molecular weight, treatment and temperature influence on protein digestibility. The authors' analysis is interesting, and their findings can provide data reference for creating foods with fast or slow digestion of protein. I think it can be published in this journal after minor modifications.

The following are the relevant questions in the article:

1)Please revise the Table in the manuscript according to the 3-line format.

2)Why does the author not use multivariate statistical analysis to study the HT at different characteristics, so as to clearly explain the influence on HT.

Author Response

Thank you!

  1. We have updated all tables according to the reviewer's instruction.
  2. PLS was used because we had one dependent variable, and PLS allowed us to model the underlying information of those variables (both dependent and independent variables).

Reviewer 2 Report

Molecules

Analysis of the factors affecting in vitro pepsinolysis of food proteins

The article intended as a collection of data might be interesting. Discursive and easy to understand. The only doubt is on Figure 1 that, as described in the figure legend has been adapted from another article. It should be improved, however, since a cut off appears on part C.

Author Response

Thank you!

  1. We have improved Figure 1 according to the reviewer's kind suggestions to include references to all five mentioned protein forms.
  2. Yes, part of the Figure (casein structure) is adapted, it was completely re-drawn by one of us. This is shown in detail in the attached document (please see attachment). 

Reviewer 3 Report

The authors of this paper performed a meta-analysis to elucidate the most important factors affecting static in vitro protein digestion. The study was well conducted, and data are presented in a clear way. Also, the discussion is thorough and rather complete. I only have a few, minor comments that must be addressed before the paper can be published.

Title: add “static” to in vitro pepsinolysis

Make clearer in L95 which milk types are considered.

Add all five protein forms in figure 1.

Define “protein half-time”

While mentioning plant proteins in the introduction, they were not clearly present/discussed in the body of the paper. Can you comment on that?

Author Response

Thank you!

  1. The title has been updated to include the word "static".
  2. Line 95 has been updated to include information that the dominant milk type of bovine milk. However, two papers also include human and equine milk (discussed in L227)
  3. Figure 1 has been updated to add a depiction of 'gel', and the caption has been updated to make clear reference to 'native' protein
  4. Half time has now been clearly defined in Line 83. 
  5. Plant proteins (commonly in our data is sorghum flour) are typically represented in the 'real food' group of proteins. All discussions of 'real food' pertain also to those plant proteins.